# A Control Strategy to Avoid Drop and Inrush Currents during Transient Phases in a Multi-Transmitters DIPT System

Wassim Kabbara [1,2,3,*], Mohamed Bensetti [1,2], Tanguy Phulpin [1,2], Antoine Caillierez [3], Serge Loudot [3] and Daniel Sadarnac [1,2]

1 Laboratoire de Génie Electrique et Electronique de Paris, CNRS, CentraleSupélec, Université Paris-Saclay, 91192 Gif-sur-Yvette, France; mohamed.bensetti@centralesupelec.fr (M.B.); tanguy.phulpin@centralesupelec.fr (T.P.); daniel.sadarnac@centralesupelec.fr (D.S.)
2 Laboratoire de Génie Electrique et Electronique de Paris, CNRS, Sorbonne Université, 75252 Paris, France
3 Renault, 1 Avenue du Golf, 78084 Guyancourt, France; antoine.caillierez@renault.com (A.C.); serge.loudot@renault.com (S.L.)
* Correspondence: wassim.kabbara@centralesupelec.fr

**Abstract:** Electrical Vehicles (EVs) have gained popularity in recent years in the automotive field. They are seen as a way to reduce the $CO_2$ footprint of vehicles. Although EVs have witnessed significant advancement in recent years, they still have two major setbacks: limited autonomy and long recharging time. Dynamic Inductive Power Transfer (DIPT) systems permit charging EVs while driving, provide unlimited autonomy, and eliminate stationary charging time and lower battery dependency. Multiple transmitters are required to achieve DIPT; thus, dealing with transient phases is essential because every time a receiver crosses over from one transmitter to another, it experiences a new transient phase. This article presents a novel control strategy for multi-transmitter DIPT systems that ensures a continuous and stable power transfer to a moving EV. The proposed control strategy eliminates drop and inrush currents during transient phases. The control integrates a soft start feature and a degraded operating mode at a predefined maximum current value. The studied structure is a symmetrical series–series compensation network. Each transmitter coil is driven by a variable frequency inverter (around 85 kHz) to ensure Zero Phase Angle mode. The control strategy was numerically validated using MATLAB Simulink and then tested experimentally. Results show a relatively low power disruption after applying the proposed control during transmitter sequencing.

**Keywords:** Dynamic Inductive Power Transfer; frequency control; series-series compensation network; voltage copying; state machine

## 1. Introduction

The future of mobility tends towards reducing, or even eliminating at some point, the use of classic fuels (petrol, diesel) to minimize the immense pollution they cause [1]. Some cities have already set strict regulations prohibiting vehicles of specific type and age from circulating on their roads, such as implementing the CRIT'Air certificate in France [2]. Therefore, car manufacturers are obliged to adapt quickly to meet the strict regulations set for internal-combustion engine (ICE) vehicles. Car electrification is a possible alternative that most car manufacturers opt for and try to put into service [3]. Hybridization is a form of vehicle electrification that combines ICE technology with a battery-powered electric motor. The most common vehicle hybridization is known in the form of Hybrid Electrical Vehicles (HEV) and Plug-in Hybrid Electrical Vehicles (PHEV). It is a well-matured technology that reduces the carbon footprint of vehicles, but its major downside is that it stays dependent on fossil fuels [4]. Another good candidate is hydrogen fuel cell technology since hydrogen has a much higher energy density (33 kW/kg) compared to today's batteries (0.2 kW/kg) [5]. However, several drawbacks significantly limit the deployment of hydrogen technology: complex onboard hydrogen storage systems are limited at 1.2 kWh/liter at 700 bars [6] and

low fuel cell efficiency (around 50%) combined with poor reliability and short lifetime [7]. One of the strongest candidates for replacing ICE vehicles are Electrical Vehicles (EVs). However, EVs have two significant drawbacks that prevent them from being the perfect alternative: limited autonomy and extended recharging time.

Lately, Dynamic Inductive Power Transfer systems (DIPT) have gained momentum. They consist of multiple transmitting coils embedded in the road, sending energy by magnetic induction to an embedded receiver coil inside the EV, thus charging wirelessly while in motion. DIPT systems can provide unlimited autonomy, eliminate stationary charging time, and lower battery dependency. It is being introduced to solve EVs' drawbacks by offering an unlimited range on the roads equipped with the DIPT system [8–10]. DIPT systems have been studied and proven to have great potential by numerous publications in the literature [11–13]. However, we have noticed a common problem concerning power interruption due to transient phases when the secondary coil passes from one primary coil to another. This problem was apparent in the results shared by the FABRIC European project that implemented a whole prototype DIPT infrastructure, including demonstrations of inductive wireless power transfer in different real driving conditions (up to 20 kW, from 0 to 100 km/h) [14]. The consequence of the power interruption during transient phases is a periodic pulse profile charging current, perturbing the Li-ion battery performance and reducing the average charging power compared to a constant current profile. A study presented in [15] showed that the profile performance index, an indicator of the quality of a current profile, is affected detrimentally both in the charge and in the discharge direction in the case of a periodic pulse current profile. One possible solution for lowering the impact of transitioning phases is by lowering the number of transitions from one ground coil to another by using a long ground coil as in [16]. Other propositions have been made to use smaller ground coils and permanently energize them with proper synchronization to modulate the power and avoid transitioning phases during movement [17,18]. However, the solutions proposed in [16–18] cannot comply with the ICNIRP [19] recommendations due to having energized ground coils radiating magnetic fields without the presence of a shielding-equipped vehicle. Another issue studied by [20], but not as widely treated in the literature, is how to control inrush currents in inductively coupled power systems. Limiting the inrush currents is critical during the start-up phase of the power transfer, especially in the presence of DC-link capacitors in the receiver system. Multiple transmitters are required to achieve DIPT; thus, dealing with transient phases is essential because every time a receiver crosses over from one transmitter to another, it experiences a new transient phase.

This article presents an original control strategy for the primary system that ensures a continuous power transfer to a secondary system in a series–series (S-S) compensation topology with multi-transmitters. The proposed control strategy eliminates drop and inrush currents during transient phases. Moreover, a soft-start algorithm and a degraded mode feature were implemented in the control algorithm. The proposed control was validated numerically using MATLAB Simulink on a model that contains four primary coils and one secondary coil of identical size separated by a 15 cm air gap. Simulation results show that the system transfers 1 kW of power with a relatively low power disruption when sequencing to a secondary coil coupled with a resistive load. Moreover, experimental testing of the presented control was done at 700 W with satisfying results.

## 2. Topology of the Studied System

### 2.1. Magnetic Coupler Architecture

The ground and secondary coils' adopted geometry was 48 × 48 cm square with six turns. Ferrite plates were added to the coils to canalize the magnetic flux better, thus enhancing the coupling coefficient (Figure 1 and Table 1).

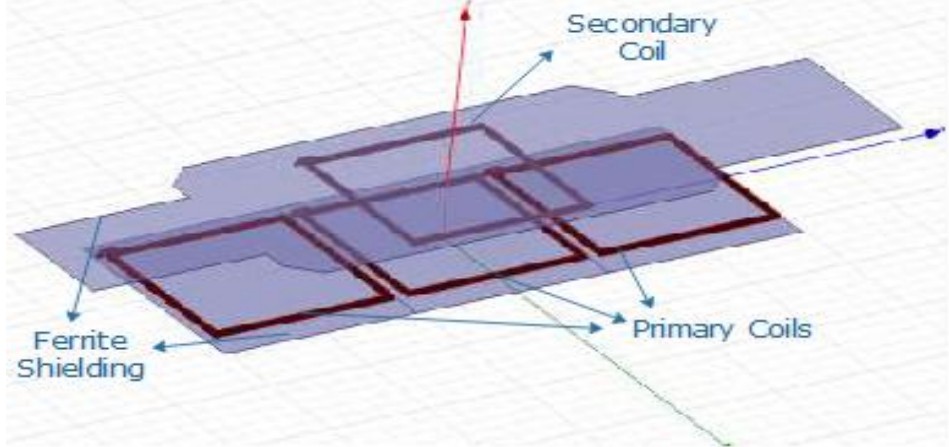

**Figure 1.** Model of the implemented coils + Ferrite plates for shielding.

**Table 1.** Characteristics of the used coils.

| Parameter | Value | Unit |
|---|---|---|
| Number of turns | 6 | turns |
| Layout | 2 layers of 3 | turns |
| Space between turns | 0 | mm |
| Space coil/ferrite | 10 | mm |
| Exterior diameter of the cable | 5 | mm |
| Total thickness (coil + ferrite) | 22 | mm |
| Cable length per coil | 10.44 + 1.5 for connections | m |
| Measured inductance (without the effect of secondary ferrite) | 45 | μH |
| Measured inductance (with the secondary placed centered above the primary) | 65 | μH |
| Air gap (center to center) | 15 | cm |

### 2.2. Power Electronics Architecture

The power electronics architecture used in this study was an H-bridge on the primary side connected to a symmetrical series compensation network and a diode bridge on the secondary side, as shown in Figure 2.

### 2.3. Electrical Equations of the Symmetrical Series Compensation Network

The reduced electrical model of the system, between the output of the inverter and the load, is presented in Figure 3, while the parameters' definitions are given in Table 2. The first-order harmonic approximation is considered to obtain the equations of the system.

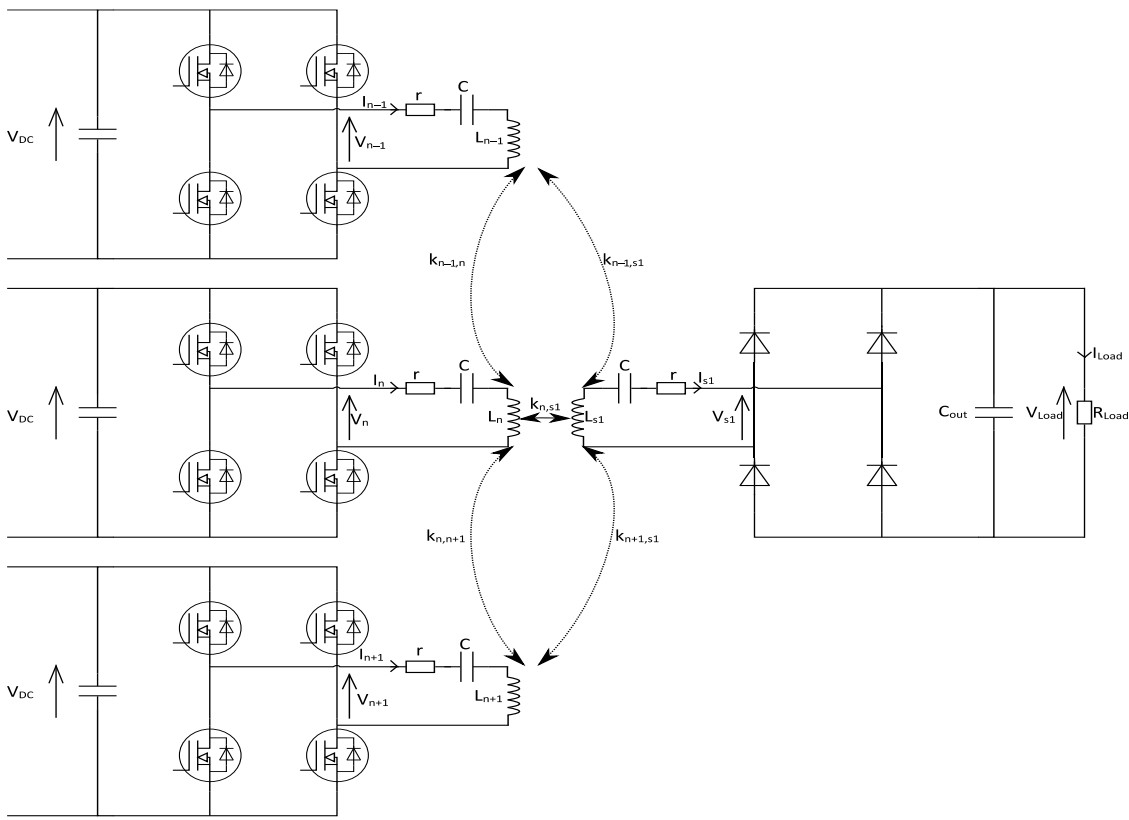

**Figure 2.** System's equivalent model with three transmitters and one receiver.

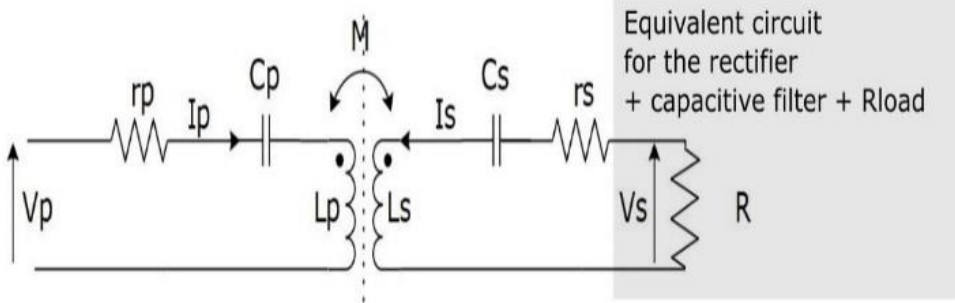

**Figure 3.** Coupler model with series–series compensation with one transmitter and one receiver.

**Table 2.** Parameter's definition.

| Parameters | Definition |
| --- | --- |
| $V_p$ | Voltage applied on the primary side |
| $r_p$ | Total equivalent resistance on the primary side |
| $C_p$ | Primary series compensation capacitor |
| $L_p$ | Inductance of the primary coil |
| $M$ | Primary to secondary mutual inductance |
| $k_{i,j}$ | Magnetic coupling between $coil_i$ and $coil_j$ |
| $L_s$ | Inductance of the secondary coil |
| $C_s$ | Secondary series compensation capacitor |
| $r_s$ | Total equivalent resistance on the secondary side |
| $V_s$ | Voltage across the equivalent load |
| $R$ | Equivalent circuit for the rectifier + capacitive filter + $R_{load}$ |
| $C_{out}$ | Output capacitance |

The following equations are developed based on the circuit provided in Figure 3:

$$V_s = -RI_s \tag{1}$$

$$M = k\sqrt{L_p L_s}, \text{where } k \text{ is defined as the coupling coefficient} \tag{2}$$

$$G_v = \frac{V_s}{V_p} \tag{3}$$

$$\overline{V} = \begin{pmatrix} V_p \\ 0 \end{pmatrix}; \overline{I} = \begin{pmatrix} I_p \\ I_S \end{pmatrix}; \overline{V} = \overline{\overline{M}}_R \overline{I} \tag{4}$$

$$\overline{\overline{M}}_R = \begin{pmatrix} r_p + L_p w j + \frac{1}{jC_p w} & k\sqrt{L_p L_s} jw \\ k\sqrt{L_p L_s} wj & R + r_s + jL_s w + \frac{1}{jC_s w} \end{pmatrix} \tag{5}$$

$$\overline{I} = \overline{\overline{M}}^{-1} \overline{V} \tag{6}$$

Identical primary and secondary coils with identical compensation capacitors are chosen to obtain the simplest analytical design. Frequency bifurcation is a phenomenon studied in [21–24] and results from having symmetrical primary to secondary systems. A fixed voltage gain can be obtained at a specific frequency in S-S compensation, as shown in [24]. This frequency exists and is not unique (Equation (11)) and is presented under specific power constraints (Equation (17)).

$$C_p = C_s = C \; ; L_p = L_s = L \; ; r_p = r_s = 0 \tag{7}$$

$$\overline{I} = \begin{pmatrix} I_p \\ I_s \end{pmatrix} = \begin{pmatrix} \frac{CwV_p(-CLw^2 + jCRw + 1)}{jC^2L^2k^2w^4 - jC^2L^2w^4 - RC^2Lw^3 + jCLw^2 2 + RCw - j} \\ \frac{C^2LV_p kw^3}{jC^2L^2k^2w^4 - jC^2L^2w^4 - RC^2Lw^3 + jCLw^2 2 + RCw - j} \end{pmatrix} \tag{8}$$

$$P_{out} = RI_s{}^2 = V_p{}^2 RC^4 L^2 w^6 * \frac{1}{\alpha + \beta^2} \\ \begin{cases} \alpha = \left[ C^2 L^2 w^4 (1 - k^2) - 2CLw^2 + 1 \right]^2 \\ \beta = RCw(1 - CLw^2) \end{cases} \tag{9}$$

$$Z_{in} = \frac{V_p}{I_p} = Re\{Z_{in}\} + Im\{Z_{in}\} * j \begin{cases} \varphi_p = arctang\left( \frac{Im\{Z_{in}\}}{Re\{Z_{in}\}} \right) \\ Re\{Z_{in}\} = -\frac{C^2 L^2 R k^2 w^4}{C^2 L^2 w^4 + C^2 R^2 w^2 - 2CLw^2 + 1} \\ Im\{Z_{in}\} = \frac{(CLw^2 - 1)(-C^2 L^2 k^2 w^4 + C^2 L^2 w^4 + C^2 R^2 w^2 - 2CLw^2 + 1)}{Cw(C^2 L^2 w^4 + C^2 R^2 w^2 - 2CLw^2 + 1)} \end{cases} \tag{10}$$

$$w_{\varphi_p = 0} = \begin{cases} \frac{\sqrt{2L - CR^2 - \gamma}}{\sqrt{2}\sqrt{CL}\sqrt{1 - k^2}} \\ \frac{1}{\sqrt{CL}} \\ \frac{\sqrt{2L - CR^2 + \gamma}}{\sqrt{2}\sqrt{CL}\sqrt{1 - k^2}} \end{cases} \tag{11}$$

$$\gamma = \sqrt{C^2 R^4 - 4CLR^2 + 4L^2 k^2}$$

$$R_{\varphi_p = 0} = R_{Im\{Z_{in}\} = 0} = \frac{\sqrt{(CLw^2 + kCLw^2 - 1)(-CLw^2 + kCLw^2 + 1)}}{Cw} \tag{12}$$

$$P_{out_{\varphi_p = 0}} = \frac{CV_p{}^2 w}{\sqrt{C^2 L^2 w^4 (k^2 - 1) + 2CLw^2 - 1}} \tag{13}$$

$$w_{\frac{d(P_{out_{\varphi_p = 0}})}{d(w)} = 0} = \frac{1}{\sqrt{LC}(1 - k^2)^{\frac{1}{4}}} \tag{14}$$

$$P_{out_{\min\& \ \varphi_p=0}} = \frac{V_p^2 \sqrt{C}}{\sqrt{2}\sqrt{L}} * \frac{1}{\sqrt{\left(1-\sqrt{1-k^2}\right)}} \tag{15}$$

Using the first harmonic method, we can approximate $V_p$ by the following:

$$V_p = \frac{\hat{V}_p}{\sqrt{2}} \approx \frac{1}{\sqrt{2}} * \frac{4}{\pi} V_{dc} \tag{16}$$

$$P_{out_{\min\& \ \varphi_p=0}} \cong \frac{8V_{dc}^2 \sqrt{C}}{\pi^2 \sqrt{2}\sqrt{L}} * \frac{1}{\sqrt{\left(1-\sqrt{1-k^2}\right)}} \tag{17}$$

This approximation is well justified when the operating frequency is close to the resonant frequency. Thus, the system's high inductive nature will naturally filter out all higher-order harmonics at high frequencies. (See Figure 4a).

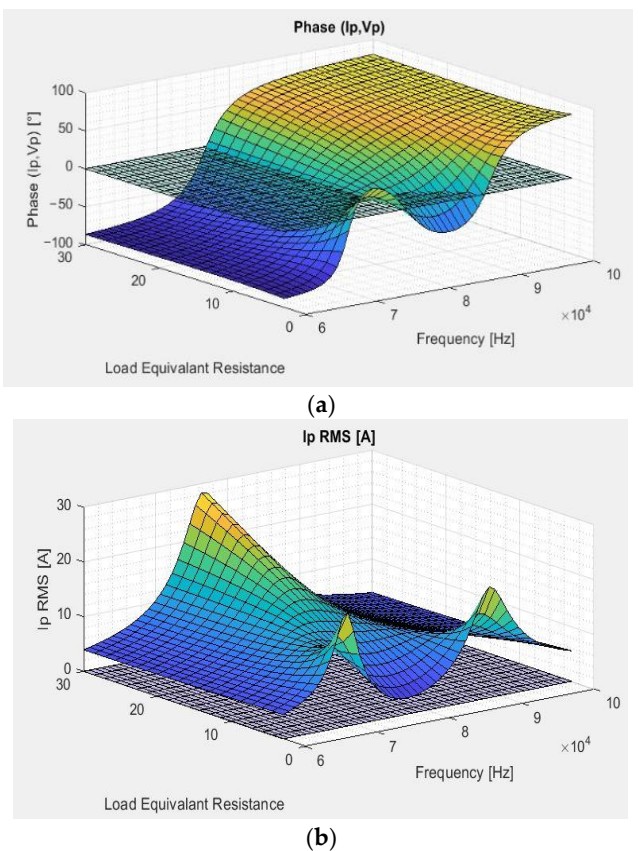

**Figure 4.** Simulation results with $C_p = C_s = 66$ [nF]; $L_p = L_s = 65$ [µH]; $r_p = r_s = 60$ [mΩ]; $V_{dc} = 60$ [V]; $k = 0.25$: (**a**) phase angle $\varphi_p$ in degrees between $I_p$ and $V_p$; (**b**) RMS current value $I_p$ in amperes.

## 3. Control Strategy of the Transmitters' Inverters

### 3.1. Control Loop

The control loop needs to comply with fast magnetic coupling variations. Besides, we wanted to reduce the complexity of the system realization by not communicating between the primary and secondary systems. Therefore, the choice was made to regulate the phase $\varphi_p$ between $I_p$ and $V_p$ by imposing the frequency $f_p$ of $V_p$, as shown in Figure 5. The power transferred to the secondary would be controlled by acting on the equivalent impedance of the secondary side.

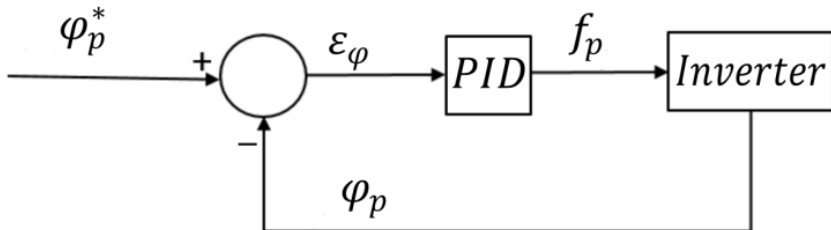

**Figure 5.** The control loop of the primary system.

Equation (17) shows that for a given electrical dimension of the system, it is possible to operate at zero phase $\varphi_p$ if the transferred power is above the minimum value $P_{out_{\min\& \varphi p=0}}$. However, Equation (11) and Figure 4a show that there are multiple solutions possible for $\varphi_p = 0$. Fixed frequency solution at $f_0 = \frac{1}{2\pi\sqrt{LC}}$ is abandoned for two main reasons:

1. Primary to secondary voltage gain ($G_v$) varies considerably with the variation of $k$ and $P_{out}$. It reaches dangerously high levels with low power transfer;
2. Impossible to operate without the presence of the secondary system or with low coupling values since the primary coil will act as a short circuit, and a substantial current $I_p$ will circulate.

The adopted solution operates at the highest frequency that satisfies $\varphi_p = 0$. It has been shown in [25,26] that operating at this resonant frequency gives the system a unique property named "Voltage Copying", where $G_v$ is very close to 1 (Figure 6). Hence, at normal operating conditions, the control of the primary system insures $\varphi_p = 0$, while power regulation is made on the secondary side with $G_v \approx 1$ and under the condition $P_{out} \geq P_{out_{\min\& \varphi p=0}}$.

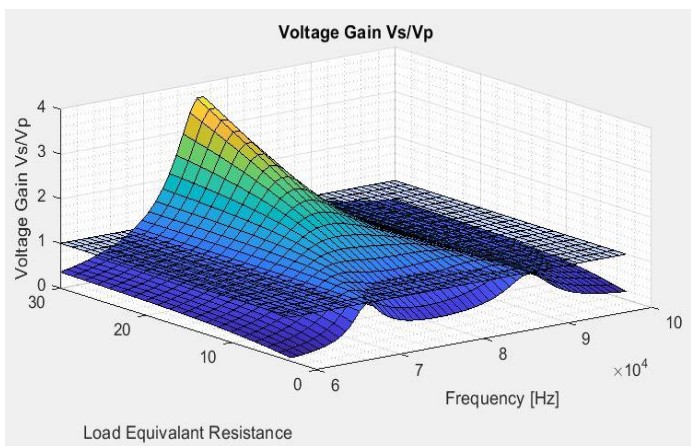

**Figure 6.** $G_v$ Simulated with $C_p = C_s = 66[\text{nF}]$; $L_p = L_s = 65[\mu\text{H}]$; $r_p = r_s = 60[\text{m}\Omega]$; $V_{dc} = 60 [\text{V}]$; $k = 0.25$.

However, power limitation via the control of the primary system is also achievable. This control could be done by simply increasing the frequency $f_p$ and thus increasing the input impedance $Z_{in}$ and limiting the current $I_p$ (check Figure 4b). In this case, $I_p$ becomes limited, but $\varphi_p$ will no longer be maintained at zero. The transmitted active power will, therefore, decrease. This property will be used in Section 3.2 to integrate a soft-start algorithm and a degraded mode using the control of the primary system.

### 3.2. Soft-Start and Degraded Mode Integration

A soft-start algorithm was implemented to avoid the current inrush during system start-ups. Another feature we implemented using the proposed control was the power degradation mode under certain conditions. As explained in Section 3.1, we did not impose

a specific power transfer from the ground side because the vehicle controls the power demand. Still, we can limit it by increasing the input impedance from the primary side.

During the start-up phase, we set $\varphi_p^*$ for $80^\circ$ (high impedance $\to$ low power transfer $\to$ controlled current inrush). Then, we decreased $\varphi_p^*$ gradually, using a step = "step_down" until reaching the defined minimum value $\varphi_p^* = 0^\circ$. Thus, enabling functioning at "Voltage Copying" mode as described previously. Power transferred will increase gradually until reaching the nominal power at $\varphi_p^* = 0^\circ$. This power corresponded to the nominal power decided by the secondary side and should respect a minimum value stated by Equation (18). The minimum duration of the soft-start phase was linked to the values of "step_down" and the execution time ($t_{exec\_cycle}$) of one cycle in the control loop.

$$t_{soft\_start_{min}} = t_{exec\_cycle} * \frac{maximum\ phase\ setpoint}{step\_down} \tag{18}$$

However, the actual total time will depend on the rapidity of the system since the algorithm does not update the value of $\varphi_p^*$ until the actual measurement of $\varphi_p$ reaches the old setpoint (Figure 7).

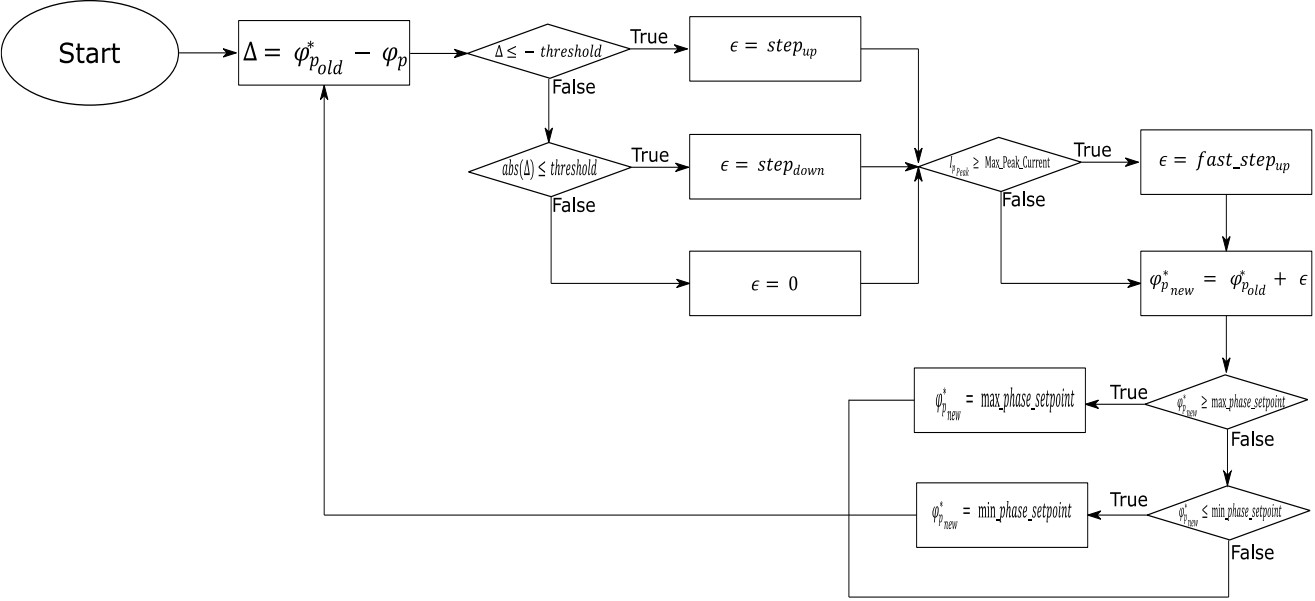

**Figure 7.** Dynamic set point implementation of $\varphi_p^*$ algorithm.

The degraded mode is achieved by limiting the value of $I_p$ and by varying the impedance value. Whenever $I_p$ reaches a predefined critical value "Max_Peak_Current", the value of $\varphi_p^*$ is increased, using a step = "fast_step_up", consequently increasing the input impedance $Z_{in}$ and limiting $I_p$ (Figure 7).

### 3.3. Sequencing Control

The proposed solution is based on the hypothesis that an independent microcontroller controls each inverter, and each microcontroller can only communicate with the neighboring's inverters, as shown in Figure 8.



**Figure 8.** Communication pathways between microcontrollers.

Each inverter follows a specific sequencing to provide a continuous and robust power flow from the ground to the vehicle during transitioning from one coil to another. The state machine of each inverter is provided in Figure 9, with descriptions of each state provided in Table 3.

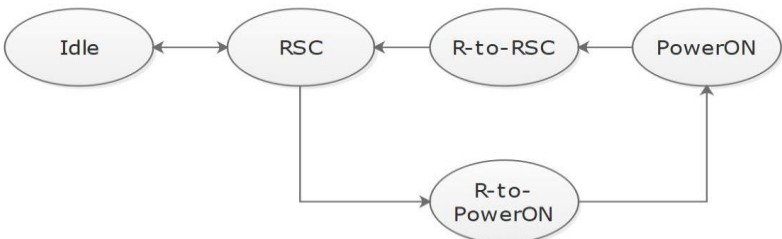

**Figure 9.** The state machine of $Inv_n$.

**Table 3.** States description.

| State | Description |
|---|---|
| Idle | All transistors of the H-bridge are set to OFF |
| RSC | The lower transistors of the H-bridge are set to ON. This way, the ground coil is in a Resonant Short Circuit with the capacitor in series |
| PowerON | The H-bridge is controlled using the dynamic phase set point to transfer power |
| R-to-PowerON | The inverter would be in RSC mode, but ready to pass to PowerOn |
| R-to-RSC | The inverter would be in PowerON mode, but ready to pass to RSC |

A step-by-step demonstration of a transition from $coil_n$ to $coil_{n+1}$ is presented in Figure 10. By checking Figure 10, a geometrical symmetry could be noticed when transitioning from $coil_n$ to $coil_{n+1}$. Under the conditions presented in Section 3.1, the system also showed electrical symmetry by having $I_{p_n} \approx I_{p_{n+1}}$. This feature has been presented with details in [26]. The key to maintaining a constant power transfer is to create a temporal symmetry around the transition. In other terms, the global system's state (inverters' states, PID memory, internal variables used in the control loop, etc.) at the instant of transition ($t_0$), which corresponds to the instant of geometrical and electrical symmetry, should be identical to the global system's state after the execution of the sequencing steps. Hence, at $t_0$, the control variables' values were saved and transferred from $microcontroller_n$ to $microcontroller_{n+1}$.

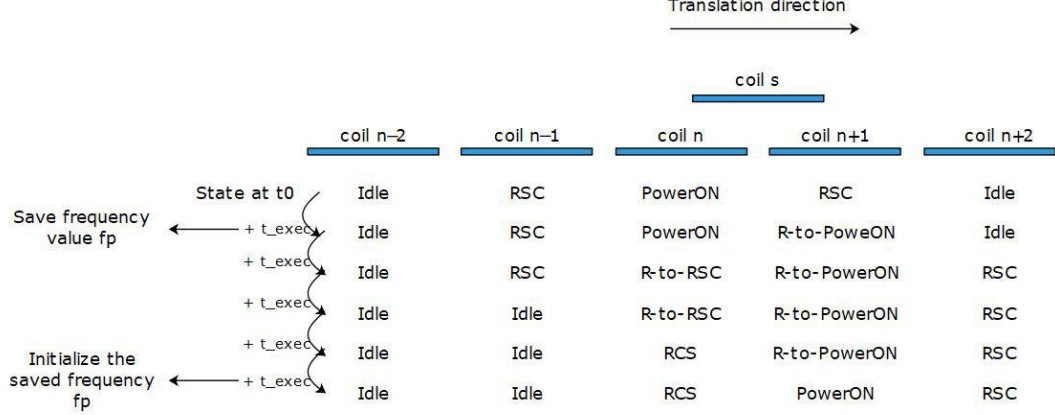

**Figure 10.** A step-by-step execution of a transition from $coil_n$ to $coil_{n+1}$.

## 4. Simulation Results

The general block diagram of the MATLAB Simulink electrical model is presented in Figure 11. A DC source was put inside the "Power source" block, four H-bridges with the

control loop and PWM generation were put inside the "Ground Side Module" block, the compensation network with the coils' model was put inside the "Coils and compensation network" block, and finally, a rectifier alongside a resistive charge was used on the receiver side. Detailed blocks can be found in Appendix A.

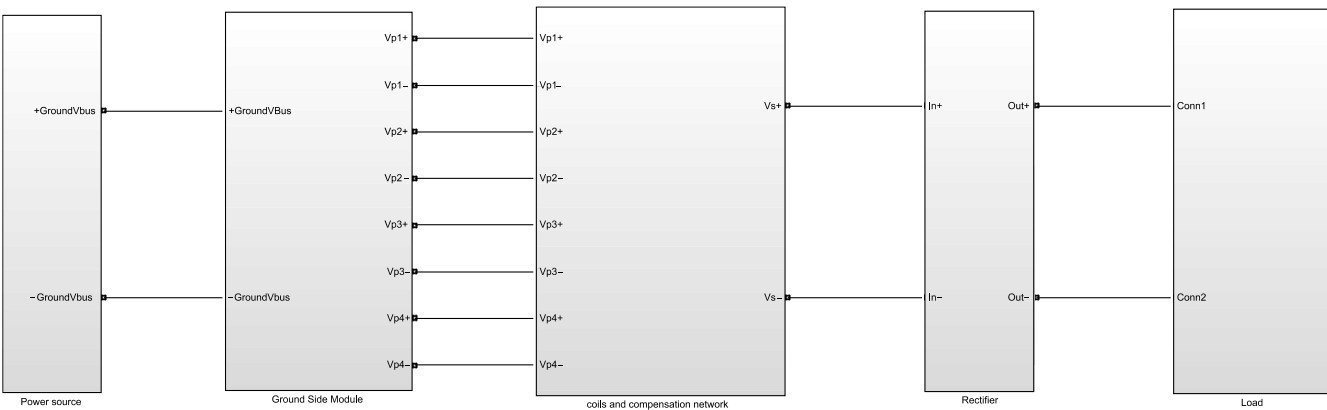

**Figure 11.** General block diagram of the MATLAB Simulink electrical model.

Table 4 presents the considered parameters in the simulation. Figures 12a and 13a present the system's behavior during the start-up phase without using a soft start. In comparison, Figures 12b and 13b show the system's behavior during the start-up phase by integrating the proposed soft-start algorithm discussed in Section 3.2. Figures 12 and 13 have been simulated using the parameters presented in Table 4.

**Table 4.** System's parameters.

| Variable | Value | Unit |
|---|---|---|
| $C_p = C_s$ | 66 | nF |
| $L_p = L_s$ | 65 | µH |
| $r_p = r_s$ | 60 | mΩ |
| $V_{dc}$ | 60 | V |
| $k_{max}$ | 24 | % |
| $k_{min}$ | 14 | % |
| $R$ | 6 | Ω |
| $C_{out}$ | 300 | µF |
| Threshold | 4 | ° |
| Step_down | 0.4 | ° |
| Step_up | 0.1 | ° |
| Fast_step_up | 4.5 | ° |
| Max_phase_stepoint | 80 | ° |
| Min_phase_stepoint | 1 | ° |
| $F_{DSP}\left(t_{exec\_cycle}\right)$ | 15 (66) | kHz (µs) |
| $P$ | 0.01 | Proportional parameter in parallel form |
| $I$ | 350 | Integrator parameter in parallel form |
| D | $10^{-4}$ | Derivative parameter in parallel form |
| $F_{init}$ | 100 | kHz |
| $F_{min}$ | 82 | kHz |
| $F_{max}$ | 100 | kHz |
| $V_{speed}$ | 25 | km/h |

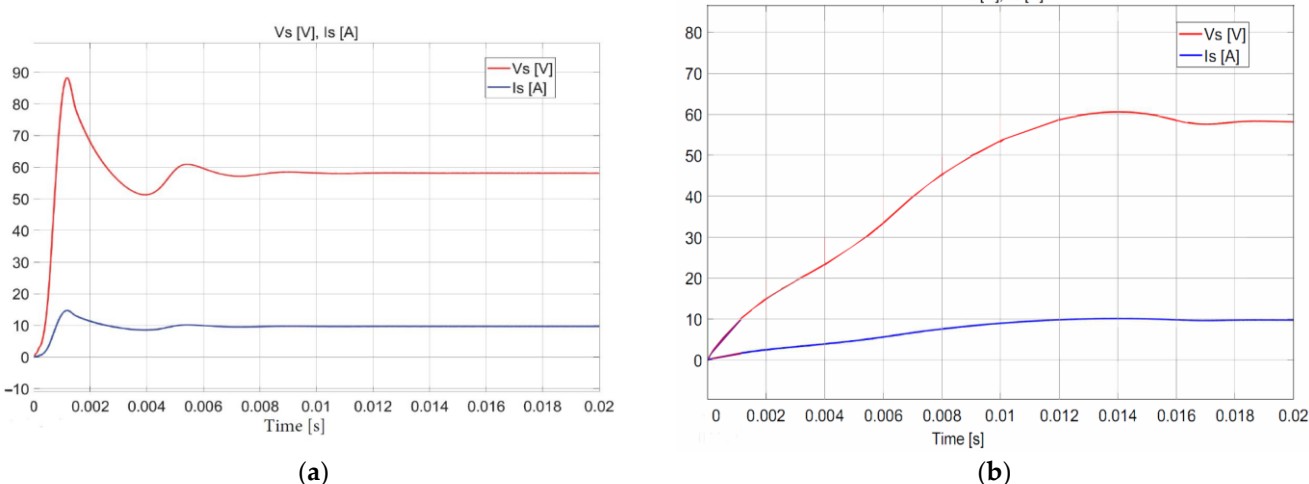

**Figure 12.** Values of $V_s$ in volts & $I_s$ in amperes: (**a**) without soft-start algorithm integrated; (**b**) with soft-start algorithm integrated.

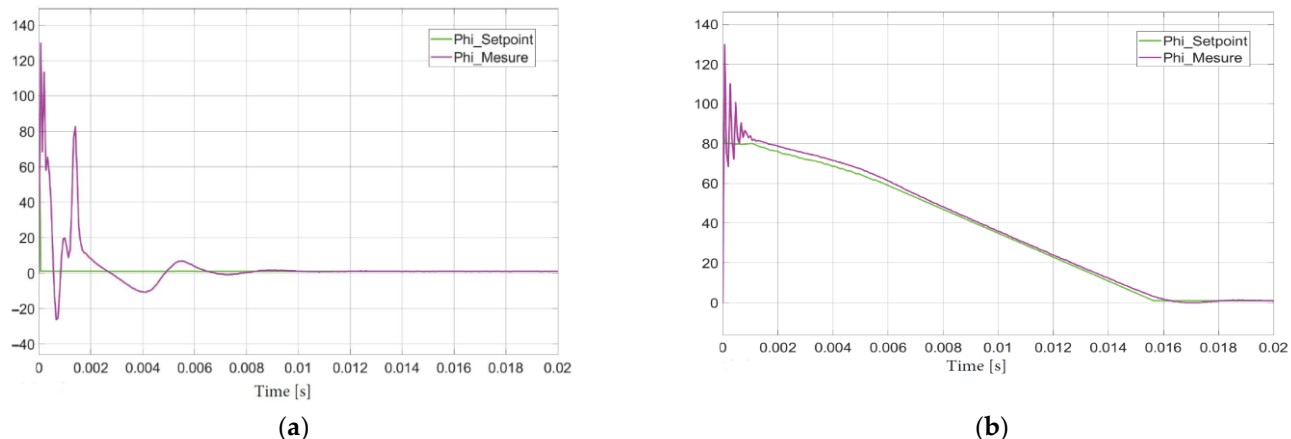

**Figure 13.** Values of $\varphi_p$ & $\varphi_p^*$ [°]: (**a**) without soft-start algorithm integrated; (**b**) with soft-start algorithm integrated.

In Figure 14a, we show the value of $I_p$ with and without the current limitation control at 15 A. As explained in Section 3.2, limiting the current $I_p$ (orange curve in Figure 14a) can be achieved by acting on the value of $\varphi_p$, which is clearly shown in Figure 14b (orange curve). Whenever $I_p$ exceeds the predefined maximum value (15 A in this simulation), the system automatically reacts based on the implemented dynamic setpoint algorithm (refer to Figure 7) and increases $\varphi_p^*$. As a result, the closed-loop control will output a higher frequency increasing the equivalent impedance at the inverter's output and thus decreasing $I_p$. As long as the measured value of $\varphi_p$ is following the dynamic set point $\varphi_p^*$, and without any current limitation, the system will always try to decrease $\varphi_p^*$ in order to reach the predefined minimum value (1° in this simulation). This behavior can be seen in the blue curves in Figure 14b.

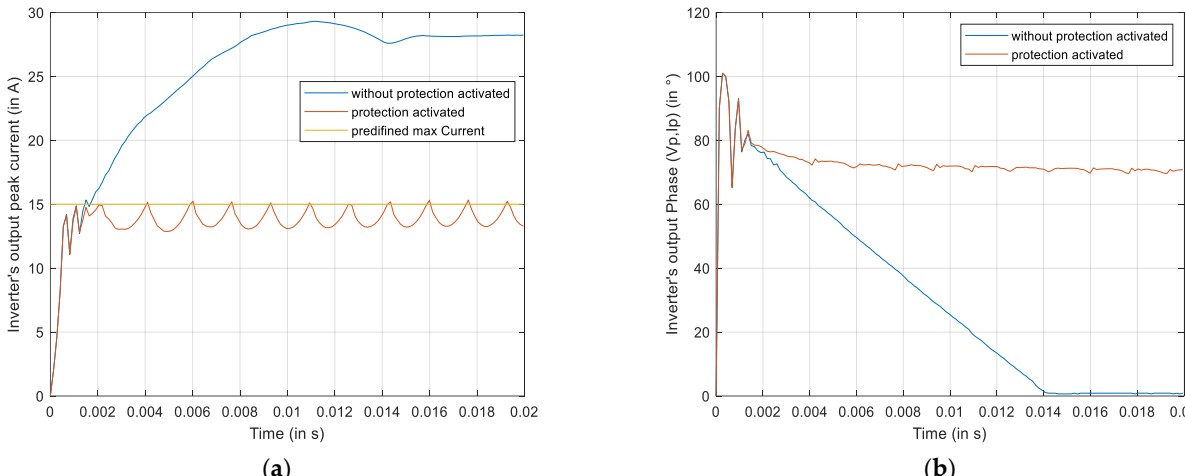

**Figure 14.** (**a**) Value of $I_p$ [A] with and without current limitation control at 15 A; (**b**) value of $\varphi_p$ [°] with and without current limitation control at 15 A.

In Figure 15a, we present the system's variables when the secondary coil passes at $V_{speed} = 25$ km/h from the center of coil$_1$ and coil$_2$ to the center of coil$_3$ and coil$_4$ without performing the proper initialization at the moment of sequencing. A noticeable power drop was located just after the sequencing from coil$_2$ to coil$_3$. This power drop is induced by not respecting the system's symmetry during the sequencing steps' execution. Another cause of this issue is the improper initializing of the PID controller of inverter$_3$. It is crucial to initialize with the exact configuration of the PID of inverter$_2$, including the memory of the integrator action. On the other hand, Figure 15b shows how the power drop due to sequencing could be reduced to negligible magnitudes when executing the proposed sequencing algorithm presented in Section 3.3. Values of the PID controller were found using the Ziegler–Nichols method as a starting point and then fine-tuned manually. The used values output the frequency directly in kHz, which explains the low values of the PID.

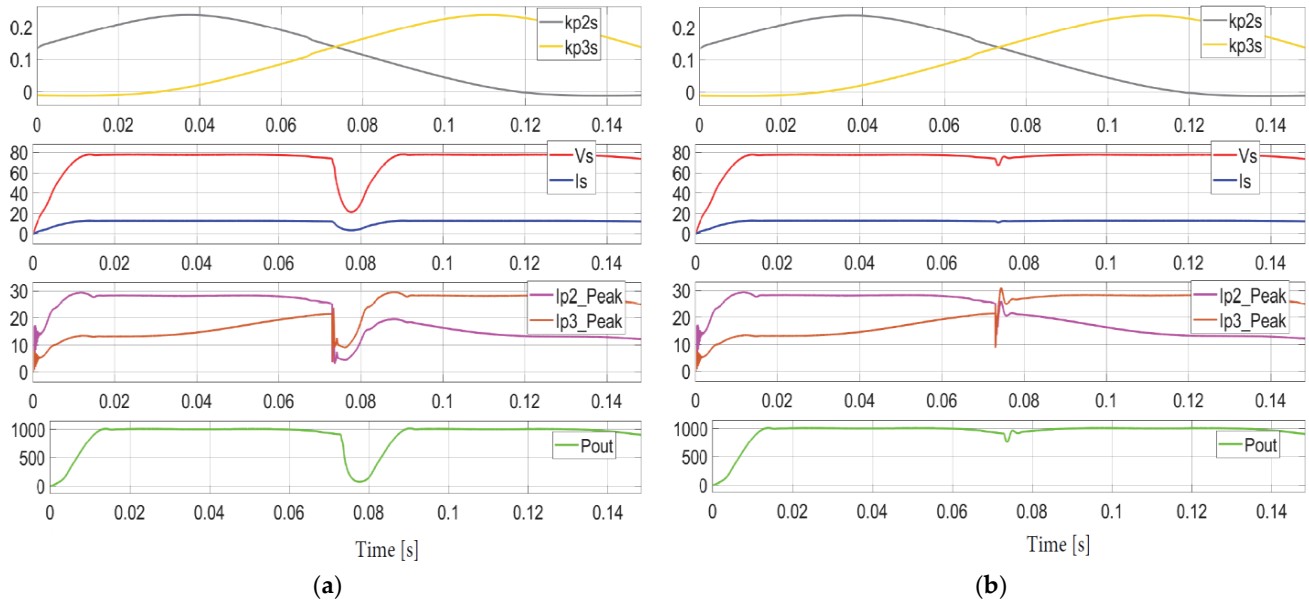

**Figure 15.** Values of $k_{p_2 s}$, $k_{p_3 s}$, $V_s$ [V], $I_s$ [A], $I_{p_2{}_{peak}}$ [A], $I_{p_3{}_{peak}}$ [A], and $P_{out}$ [W] as a function time [s] where the secondary coil is moving from the center of coil$_1$ and coil$_2$ to the center of coil$_3$ and coil$_4$: (**a**) without performing the proper initialization at the moment of sequencing; (**b**) while performing the proposed sequencing control presented in Section 3.3.

## 5. Experimental Testing

Experimental testing of the proposed control law was performed using a DIPT test bench. The block schematics of the used system are given in Figure 16a. The three inverters shared a common control card containing one DSP and one FPGA for control and monitoring. The selected DSP is the TMS320F28335 (provided by Texas Instruments, Dallas, TX, USA). It manages the system state machine, the control loop regulation, the PWM control signals, and the communication. The selected FPGA is the AGLN250V2 (provided by Microchip, Chandler, AZ, USA). It executes the voltage/current phase measurements and managed the PWM control signals sent from the DSP. The frequency of the control was set to 15 [kHz], therefore, $t_{exec\_cycle} = 66.66$ µs. The DSP was programmed using the compiled code from a MATLAB Simulink model using the Texas Instrument C2000 package in MATLAB coupled with Code Composer Studio. The chosen electrical and geometrical parameters of the test bench were identical to those defined in Tables 1 and 4. Figure 16b shows a photo of the test bench.

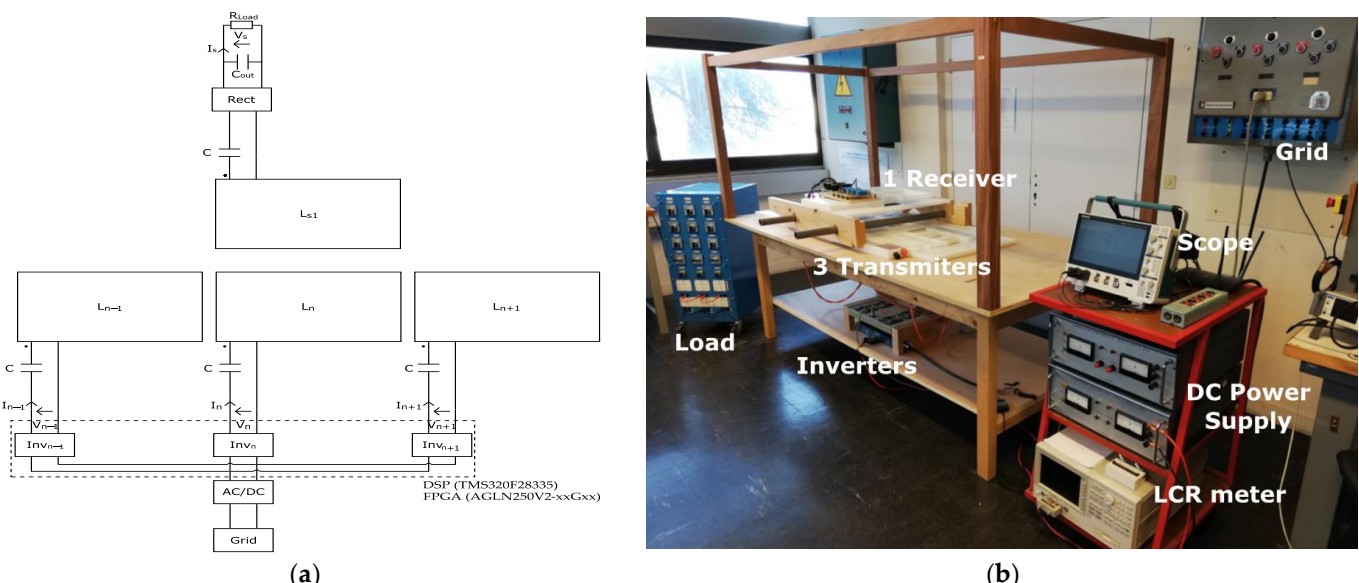

**(a)**      **(b)**

**Figure 16.** (**a**) Block schematics of the test bench; (**b**) photo of the test bench.

The oscilloscope's screen presented in Figure 17 shows the voltage and current measurements across the output of inverter$_n$ and inverter$_{n+1}$ during the transient phase from coil$_n$ to coil$_{n+1}$. It shows how there was a minimal perturbation of the $I_{p_n}$ and $I_{p_{n+1}}$ after sequencing. As we have explained earlier in Section 3.3, $I_{p_n} \approx I_{p_{n+1}}$ at the instant of sequencing due to geometrical and electrical symmetry. This explains why the currents shown in Figure 17 were nearly identical. We recall that just before the instant of sequencing, inverter$_n$ was in PowerOn mode and inverter$_{n+1}$ was in RSC mode. At the end of the sequencing steps, inverter$_n$ entered RSC mode while inverter$_{n+1}$ entered the PowerOn mode. It is clear from Figure 17 that sequencing steps induced very little perturbation to the current profile.

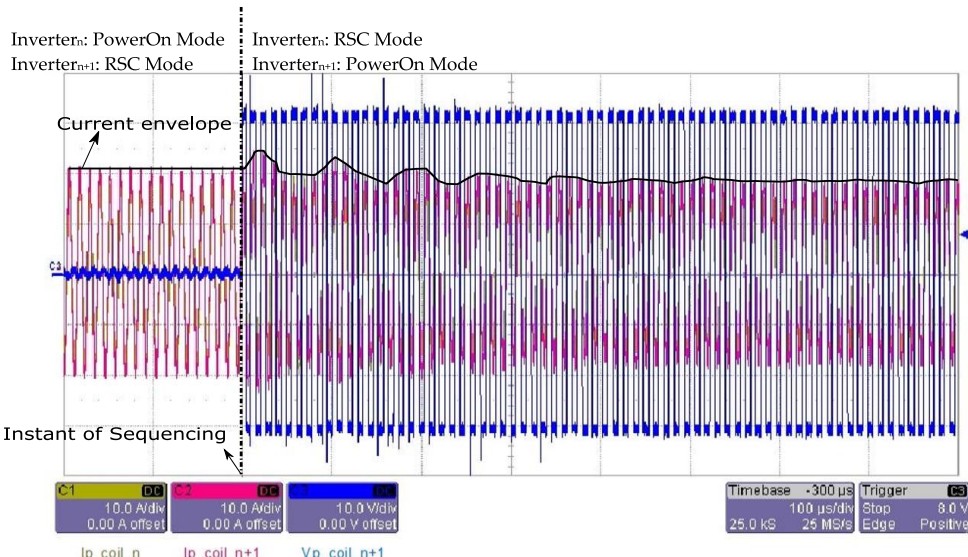

**Figure 17.** Voltage & Current measurements across the output of inverter$_n$ and inverter$_{n+1}$ during the transient phase from coil$_n$ to coil$_{n+1}$.

In Figure 18, we present the values of the output voltage ($V_s$) and the output current ($I_{out}$) as a function of time [s] where the secondary coil is moving from the center of coil$_2$ to the center of coil$_4$ while performing the proper sequencing technique presented in Section 3.3 at $t_{coil_2 \to coil_3} = 4.7$ [s] and $t_{coil_3 \to coil_4} = 8$ [s]. A minimal power drop can be noticed during the two sequencing instants.

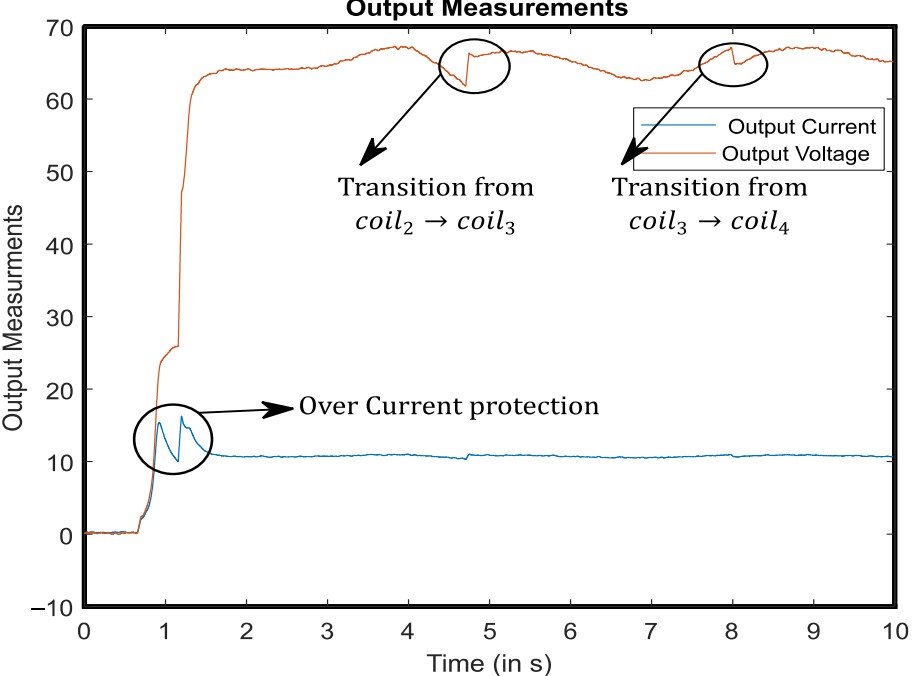

**Figure 18.** Values of $V_s$ [V] and $I_{out}$ [A] as a function of time [s] in which the secondary coil is moving from the center of coil$_2$ to the center of coil$_4$ while performing the proper sequencing technique presented in Section 3.3 at $t_{coil_2 \to coil_3} = 4.7$ [s] and $t_{coil_3 \to coil_4} = 8$ [s].

## 6. Conclusions

This paper presents a control strategy in DIPT for the primary system that ensures a continuous power transfer flow to a secondary system in a series–series compensation

topology with multi-transmitters. The proposed control strategy eliminates drop and inrush currents during transient phases. Moreover, a soft-start algorithm and a degraded mode feature were implemented in the control algorithm. The proposed control was tested numerically on the MATLAB Simulink by transferring 1 kW to a resistive load located into a moving pickup coil. Ground and pickup coils were sized identically and were separated by a 15 cm air gap. A test bench at 700 W performed preliminary experimental validations for the proposed control with satisfying results. More detailed experiments will be performed in future work with higher power and higher speeds.

**Author Contributions:** Conceptualization, W.K. and A.C.; Methodology, W.K.; Software, W.K.; Validation, M.B., T.P., D.S., A.C. and S.L.; Formal analysis, W.K.; Investigation, W.K.; Writing—review and editing, W.K.; Supervision, M.B., T.P., D.S., A.C. and S.L. All authors have read and agreed to the published version of the manuscript.

**Funding:** This research received no external funding.

**Institutional Review Board Statement:** Not applicable.

**Informed Consent Statement:** Not applicable.

**Conflicts of Interest:** The authors declare no conflict of interest. The funders had no role in the design of the study; in the collection, analyses, or interpretation of data; in the writing of the manuscript, or in the decision to publish the results.

## Appendix A

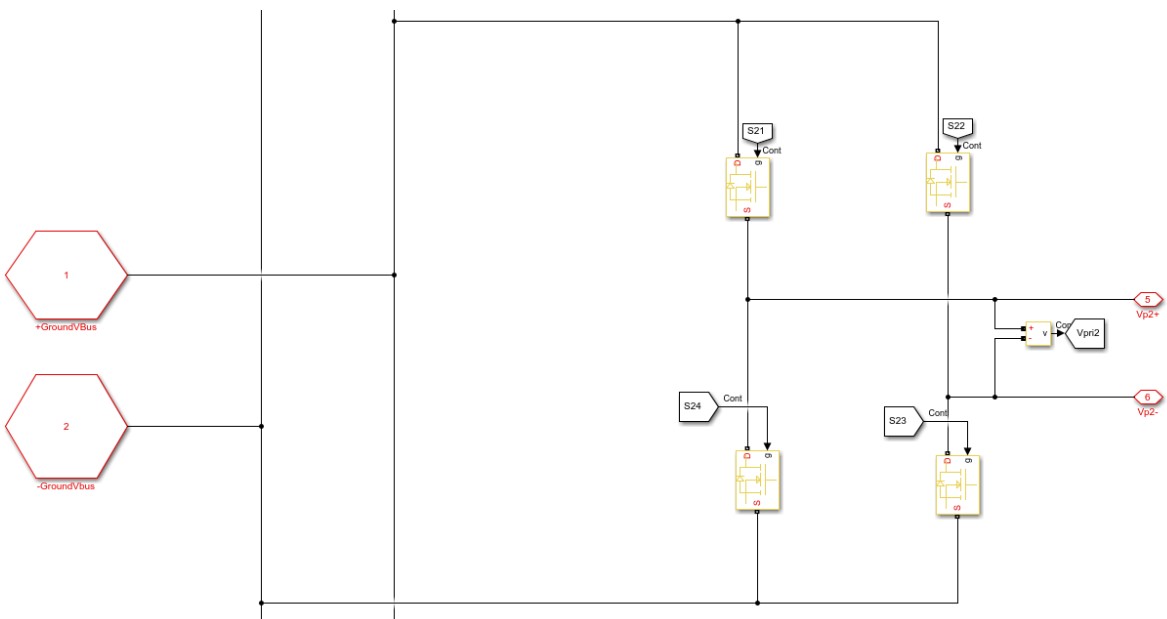

**Figure A1.** H-bridge inverter (four are used inside the "Ground Side Module" block).

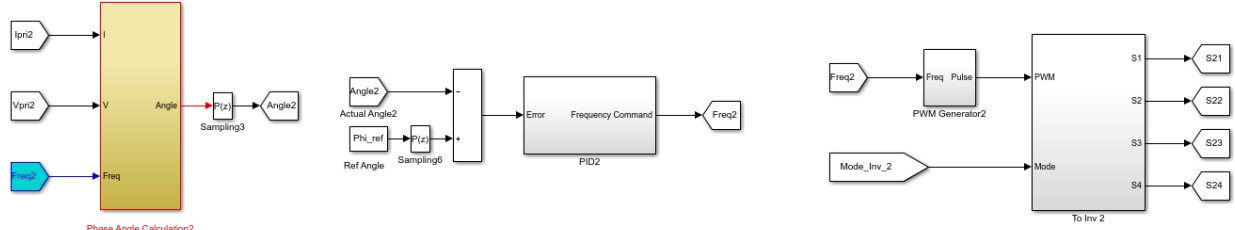

**Figure A2.** The control circuit of each inverter with PWM generation.

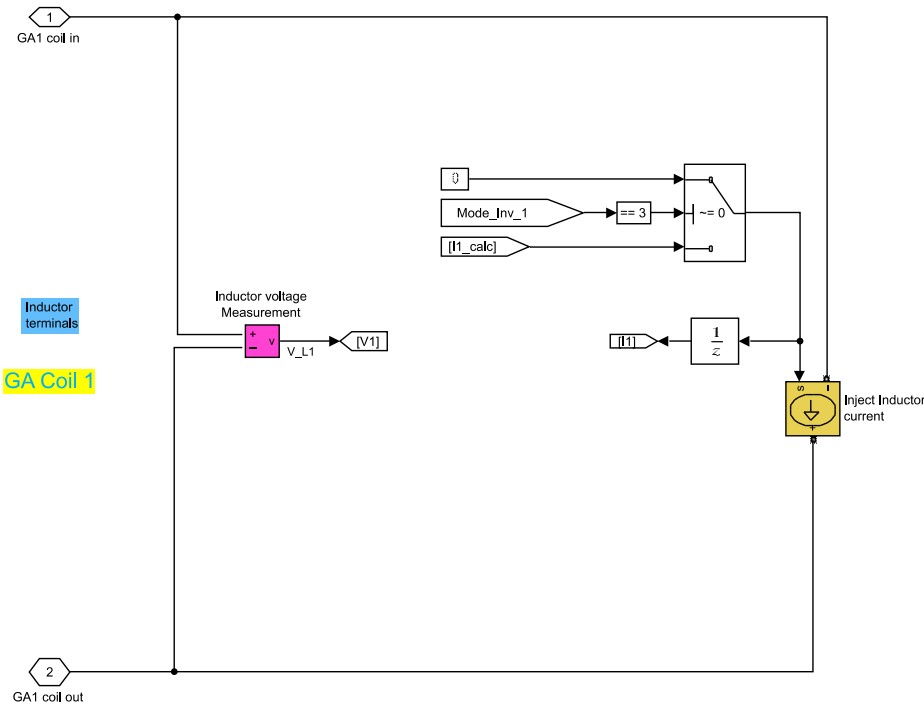

**Figure A3.** Each coil is modeled as a current source.

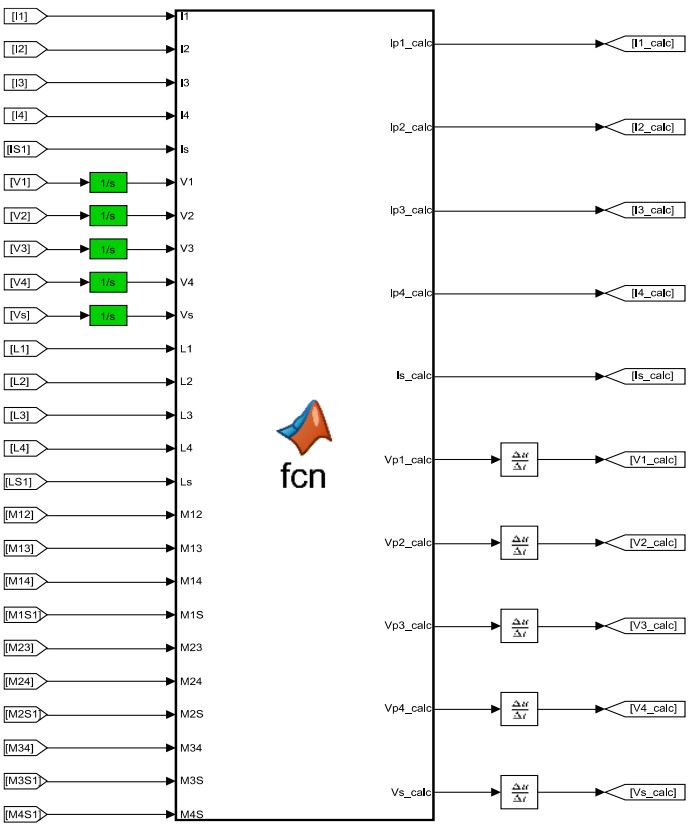

**Figure A4.** Model of the magnetic couplers.

The following electrical relations give the value of each current source:

```
function [Ip1_calc, Ip2_calc, Ip3_calc, Ip4_calc, Is_calc, Vp1_calc, Vp2_calc, Vp3_calc, Vp4_calc,
Vs_calc] = fcn(I1, I2, I3, I4, Is, V1, V2, V3, V4, Vs, L1, L2, L3, L4, Ls, M12, M13, M14, M1S, M23, M24,
M2S, M34, M3S, M4S)
% Primary Coil 1
Vp1_calc = L1*I1 + M12*I2 + M13*I3 + M14*I4 + M1S*Is; %add a derivative block at the output
Ip1_calc = (V1 - M12*I2 - M13*I3 - M14*I4 - M1S*Is)/L1; %add an integration block at input
% Primary Coil 2
Vp2_calc = L2*I2 + M12*I1 + M23*I3 + M24*I4 + M2S*Is; %add a derivative block at the output
Ip2_calc = (V2 - M12*I1 - M23*I3 - M24*I4 - M2S*Is)/L2; %add an integration block at input
% Primary Coil 3
Vp3_calc = L3*I3 + M13*I1 + M23*I2 + M34*I4 + M3S*Is; %add a derivative block at the output
Ip3_calc = (V3 - M13*I1 - M23*I2 - M34*I4 - M3S*Is)/L3; %add an integration block at input
% Primary Coil 4
Vp4_calc = L4*I4 + M14*I1 + M24*I2 + M34*I3 + M4S*Is; % add a derivative block at the output
Ip4_calc = (V4 - M14*I1 - M24*I2 - M34*I3 - M4S*Is)/L4; %add an integration block at input
% Secondary Coil
Vs_calc = Ls*Is + M1S*I1 + M2S*I2 + M3S*I3 + M4S*I4; %add a derivative block at the output
Is_calc = (Vs - M1S*I1 - M2S*I2 - M3S*I3 - M4S*I4)/Ls; %add an integration block at input
end
```

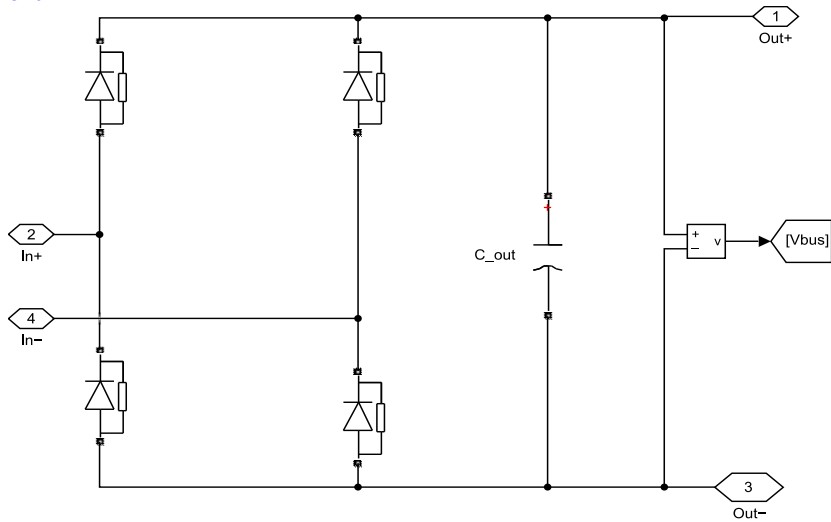

**Figure A5.** The following rectifier is used on the secondary side.

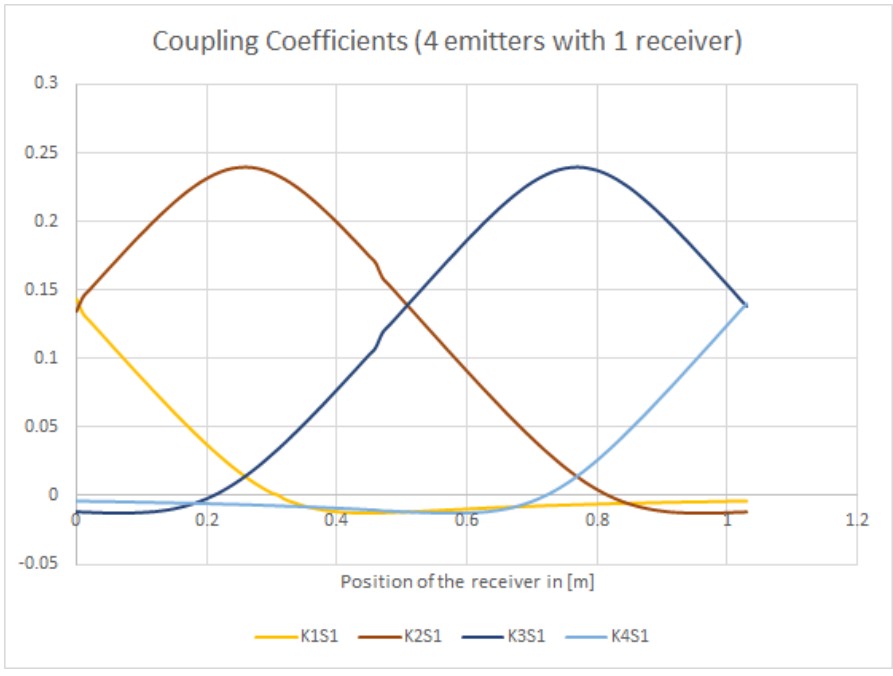

**Figure A6.** Coupling values get updated as a function of the receiver's position.

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
