# Peer review of "A Control Strategy to Avoid Drop and Inrush Currents during Transient Phases in a Multi-Transmitters DIPT System"

_energies, doi:10.3390/en15082911_

Round 1

Reviewer 1 Report

The authors need to incorporate the following suggestions for improving the technical quality of the paper.

  1. What is the impact of inrush currents on the EV batteries and what is the maximum rating of the EV Chargers and time of charging with the proposed methodology?
  2. The MATLAB Simulink electric model developed for the complete system should be included in the paper.
  3. What is the optimization technique used for tuning the PID controller parameters. The PID Controller design procedure needs to be explained.

Reviewer 2 Report

The authors present a control technique to eliminate drop and inrush currents during transient phases in Dynamic Inductive Power Transfer (DIPT) systems. The strategy is able to produce also a soft start e manage a degraded mode.

In eq. (7) you consider the two resistances of the primary and secondary equal to zero. It is a significant simplification. Can you explain what could happen if the simplification was not done?

The transitions and the ramp have to face the problem of the vehicle speed. Have you found the maximum speed? 25km/h? How to increase this value?

The experimental activities are made on the setup shown in fig. 15b. It is not clear how the problem related to the speed is considered in this test. It seems that the results are obtained ad a very low speed.

Some corrections:

Page 1, line 44: “Complex” -> “complex”

Eq. (4): IS->Is

Reviewer 3 Report

the article is interesting. It is well written. It presents both theoretical (simulations) and experimental treatment. I am satisfied from all points of view.
Authors may be asked to correct some oversights such as equation 17, the acronym E.V. which should be EV like the others, and then avoid intruding articles like "Reference [15]"

Round 2

Reviewer 1 Report

1. The MATLAB Simulink electric model developed for the complete system should be included in the final version of the paper.
